# Effect of remdesivir post hospitalization for COVID-19 infection from the randomized SOLIDARITY Finland trial

Olli P. O. Nevalainen [1,2,3,4], Saana Horstia[1], Sanna Laakkonen[1], Jarno Rutanen[2,5], Jussi M. J. Mustonen[6], Ilkka E. J. Kalliala [7,8], Hanna Ansakorpi[9], Hanna-Riikka Kreivi [10], Pauliina Kuutti[1], Juuso Paajanen [10], Seppo Parkkila[11], Erja-Leena Paukkeri[5], Markus Perola[12,13], Negar Pourjamal [1], Andreas Renner [10,14], Tuomas Rosberg[15], Taija Rutanen[16], Joni Savolainen[16], Solidarity Finland Investigators*, Jari K. Haukka [17,18], Gordon H. Guyatt[19,20] & Kari A. O. Tikkinen [21,22] ✉

We report the first long-term follow-up of a randomized trial (NCT04978259) addressing the effects of remdesivir on recovery (primary outcome) and other patient-important outcomes one year after hospitalization resulting from COVID-19. Of the 208 patients recruited from 11 Finnish hospitals, 198 survived, of whom 181 (92%) completed follow-up. At one year, self-reported recovery occurred in 85% in remdesivir and 86% in standard of care (SoC) (RR 0.94, 95% CI 0.47-1.90). We infer no convincing difference between remdesivir and SoC in quality of life or symptom outcomes ($p > 0.05$). Of the 21 potential long-COVID symptoms, patients reported moderate/major bother from fatigue (26%), joint pain (22%), and problems with memory (19%) and attention/concentration (18%). In conclusion, after a one-year follow-up of hospitalized patients, one in six reported they had not recovered well from COVID-19. Our results provide no convincing evidence of remdesivir benefit, but wide confidence intervals included possible benefit and harm.

After the acute phase of coronavirus disease 2019 (COVID-19), many survivors experience persistent symptoms, often called "long-COVID" or "post COVID-19 condition"[1,2]. The World Health Organization (WHO) defines long-COVID as symptoms continuing beyond 3 months from the onset of COVID-19 without an alternative explanation[3]. The prevalence of long-COVID varies by its definition and duration of follow-up[4]. According to a recent systematic review, up to half of patients experience some sequelae at 6 months[5]. An observational study among more than 400 hospitalized patients in Italy suggested a lower risk of long-COVID for those who received remdesivir during hospitalization (OR 0.64, 95% CI 0.41–0.78)[6]. There are, however, no published long-term follow-ups of randomized controlled trials (RCTs) on COVID-19 treatments[7].

In the randomized SOLIDARITY Finland trial, we evaluate a range of patient-relevant outcomes 1 year after hospitalization resulting from COVID-19.

## Results

### Patients

Between 23 July 2020 and 27 January 2021, we recruited 208 hospitalized patients from 11 Finnish hospitals (Fig. 1), representing 23.4% of the patients treated for more than 24 h in our study hospitals[8]. At baseline, the mean age of the patients was 58.3 years (standard deviation, SD 13.4; range 25–88 years); 64% were men, and the mean body mass index was 30.6 (SD 6.2) (Table 1). Remdesivir was started on either the day of hospital admission or the first full day of

**Fig. 1 | Study flow chart.**

**Table 1 | Characteristics of participants in a randomized controlled trial between remdesivir (RDV) and standard of care (SoC) arms at baseline and long-term follow-up after 1 year from randomization**

| | Baseline, *n* = 208 | | After 1 year, *n* = 181 | |
|---|---|---|---|---|
| | RDV *n* = 114 | SoC *n* = 94 | RDV *n* = 98 | SoC *n* = 83 |
| Age in years, mean (SD) | 57.2 (13.5) | 59.7 (13.2) | 57.7 (12.9) | 59.4 (13.0) |
| Male, *n* (%) | 74 (64.9) | 60 (63.8) | 59 (60.2) | 50 (60.2) |
| BMI, mean (SD) | 31.5 (6.35) | 29.6 (6.0) | 31.7 (6.09) | 29.9 (6.01) |
| Current smoking, *n* (%) | 2 (1.8) | 4 (4.2) | 4 (4.1) | 3 (3.6) |
| Diabetes, *n* (%) | 20 (17.5) | 16 (17.0) | 25 (25.5) | 15 (18.1) |
| **Hospital phase variables** | | | | |
| Oxygen at hospital admission, *n* (%) | | | | |
| No oxygen | 30 (26) | 20 (21) | 25 (25.5) | 15 (18.1) |
| Any oxygen | 84 (74) | 74 (79) | 73 (74.5) | 68 (81.9) |
| Treated at an intensive care unit, *n* (%) | 12 (10.5) | 11 (11.7) | 10 (10.2) | 10 (12.0) |
| Received dexamethasone, *n* (%) | 79 (69.3) | 72 (76.6) | 69 (70.4) | 61 (73.5) |
| Duration of hospitalization in days, median (IQR) | 8 (6–11) | 8.5 (6–15) | 8 (6–11) | 8 (6–14) |
| **Long-term follow-up variables, *n* (%)** | | | | |
| Working situation | | | | |
| Employed | – | – | 57 (58.2) | 40 (48.2) |
| Retired | – | – | 35 (35.7) | 34 (41.0) |
| Other (student, unemployed jobseeker, sickness allowance) | – | – | 6 (6.1) | 9 (10.8) |
| **Current ability to work (retired excluded), *n* (%)** | | | | |
| No difference compared to the time before COVID-19 | – | – | 30 (47.6) | 25 (51.0) |
| Participating in work or studies, but with deteriorated ability after COVID-19 | – | – | 33 (52.4) | 19 (38.8) |
| Unable to work or study due to symptoms associated with COVID-19 | – | – | 0 | 1 (2.0) |
| Unable to work or study due to other reasons than symptoms associated with COVID-19 | – | – | 0 | 3 (6.1) |

hospitalization in 65 patients (57%), in 36 patients (32%) on the second full day of hospitalization, and later in 12 patients (11%). The median duration of remdesivir treatment was 5 days (IQR 4–8).

**Mortality**

Of the 208 randomized patients, five (2.4%) patients died during hospitalization (due to COVID-19) and five (2.4%) during 1-year follow-up, five (2.4%) declined to participate in long-term follow-up, and we failed to reach 11 (5.3%). Of the survivors, 181 (92%) completed the 1-year survey (Fig. 1). At 1 year, 5 (4.4%) in the remdesivir and 5 (5.3%) patients in the SoC group had died (RR 0.82, 95% confidence interval (CI) 0.25–2.76; absolute difference −0.9%, 95% CI −7.9–5.3%) (Table 2).

**Recovery, symptoms, and quality of life**

Self-reported recovery (fully or largely) occurred in 85% in remdesivir and in 86% in SoC (RR 0.94, 95% CI 0.47–1.90; absolute difference −0.9%. 95% CI −11–10%). At admission, 22% did not require additional oxygen, and stratified analysis by the need of oxygen did not materially

**Table 2 | Effect of remdesivir plus standard of care (SoC) compared to SoC only on patient-relevant outcomes after 1 year from hospitalization due to COVID-19-infection**

| Outcome (Option categories in questionnaire) | Remdesivir, n = 98 (%) | SoC, n = 83 (%) | RR, 95% CI |
|---|---|---|---|
| **How do you feel you have recovered from the COVID-19 infection you had a year ago?** | | | 0.94, 0.47–1.90 |
| Fully or largely (1–2) | 83 (84.7) | 71 (85.5) | |
| About halfway recovered to not recovered at all (3–5) | 15 (15.3) | 12 (14.5) | |
| **Exertional dyspnea, mMRC dyspnea scale** | | | 0.61, 0.20–1.85 |
| No to slight dyspnea (mMRC 0–1) | 92 (93.9) | 76 (91.6) | |
| At least a need to walk slower than usually (mMRC 2–4) | 5 (5.1) | 7 (8.4) | |
| Excluded (paralyzed before COVID-19) | 1 | 0 | |
| **Fatigue** | | | 0.88, 0.54–1.44 |
| No or slight fatigue (1–2) | 74 (75.5) | 60 (72.2) | |
| Moderate or severe fatigue (3–4) | 24 (24.5) | 23 (27.7) | |
| **Mobility, walking (EQ-5D-5L)** | | | 1.03, 0.54–1.96 |
| No or slight problems (1–2) | 81 (82.7) | 69 (83.1) | |
| From moderate problems to unable to walk (3–5) | 17 (17.3) | 14 (16.9) | |
| **Self-care, washing or dressing oneself (EQ-5D-5L)** | | | 0.51, 0.13–2.08 |
| No or slight problems (1–2) | 95 (96.9) | 78 (94.0) | |
| From moderate problems to inability to wash or dress (3–5) | 3 (3.1) | 5 (6.0) | |
| **Usual activities, e.g., work, study, housework, family or leisure activities (EQ-5D-5L)** | | | 0.71, 0.32–1.55 |
| No or slight problems (1–2) | 88 (89.8) | 71 (85.5) | |
| From moderate problems to inability to do usual activities (3–5) | 10 (10.2) | 12 (14.5) | |
| **Pain or discomfort (EQ-5D-5L)** | | | 0.85, 0.44–1.63 |
| No or slight pain (1–2) | 83 (84.7) | 68 (81.9) | |
| From moderate to extreme pain (3–5) | 15 (15.3) | 15 (18.1) | |
| **Anxiety or depression (EQ-5D-5L)** | | | 1.27, 0.47–3.42 |
| No or slight problems (1–2) | 89 (90.8) | 77 (92.8) | |
| From moderate to extreme problems (3–5) | 9 (9.2) | 6 (7.2) | |

change the recovery estimate. Exertional dyspnea (at least a need to walk more slowly than usual) occurred in 5% in remdesivir and in 8% in SoC (RR 0.61, 95% CI 0.20–1.85; absolute difference −3.3%, 95% CI −12–4.4%). Median EQ-VAS was 75.5 (IQR 67.8–85.0) in the remdesivir and 80 (IQR 67.5–86.5) in SoC group (ordered logistic regression OR 0.83, 95% CI 0.49–1.40) (Table 2). We also found similar scores in the remdesivir and SoC groups in all quality-of-life domains. Regarding the 21 potential long-COVID symptoms, there were no statistically significant differences between treatment arms (Supplementary). Patients often reported moderate or major bother from fatigue (26%), joint pain (22%), persistent respiratory mucus (21%), and problems with memory (19%) and attention/concentration (18%) (Fig. 2).

## Discussion

This is the first 1-year follow-up of an RCT on COVID-19 drug treatment. We could not demonstrate long-term benefits for remdesivir in patients hospitalized due to COVID-19. After a 1-year follow-up, one in six survivors reported they had not recovered well from COVID-19. One in four reported substantial bother from fatigue. This result is similar to that reported in an Italian observational study of hospitalized patients that reported fatigue in one in three patients at 6 months post COVID-19[6].

This study has several strengths. First, we recruited one in four potentially eligible patients treated in our hospitals during the study period to our randomized trial[8,9]; this enhances the generalizability of the results. Second, we achieved a very high follow-up rate (92% of survivors). To increase participation and avoid miscommunication and misunderstanding, we translated the questionnaire into nine languages and used interpreters in phone interviews when necessary. Third, our multidisciplinary team of clinicians (representing eight

different fields), methodologists, and patient partners created a questionnaire that focused on the most patient-relevant outcomes.

The major limitation of our study is the small sample size. Adequately powered studies (with short follow-up) have earlier demonstrated that remdesivir can prevent the progression to severe disease if given in the early phase of the infection[9,10]. A fifth of our patients were not receiving oxygen at hospital admission. As the short-term benefits of remdesivir seem larger in non-severe patients[9], this could be the subpopulation of hospitalized patients who might achieve long-term benefits as well. Patients experienced much lower in-hospital mortality rates in Finland (2.4%) than in the global trial (15.0%) and were therefore a potentially more suitable patient population (likely earlier phase of the disease) for an antiviral drug[8,11]. However, our subgroup of patients in this phase was too small to inform this hypothesis. We hope that other COVID-19 treatment RCTs, especially the large trials—such as the RECOVERY and PANORAMIC, will also extend their follow-up to evaluate the potential long-term effects of acute-phase treatment both in hospitalized and non-hospitalized patients[12–14]. Finally, we did not have information regarding potential treatments post-discharge. However, as no specific treatment has been proved effective for long-COVID, this may not be a major limitation.

In conclusion, we report the first long-term results of an RCT on COVID-19 treatment. At 1-year follow-up among patients hospitalized for COVID-19, about one in four patients reported substantial bother from fatigue, and one in six felt their recovery from COVID-19 was incomplete. We could not detect any effect of remdesivir on long-term recovery, quality of life, or long-COVID symptoms, but confidence intervals are wide, including substantial benefit and harm. Determining acute-phase treatment impact on long-term sequelae of COVID-19 will require additional RCTs with adequate follow-up.

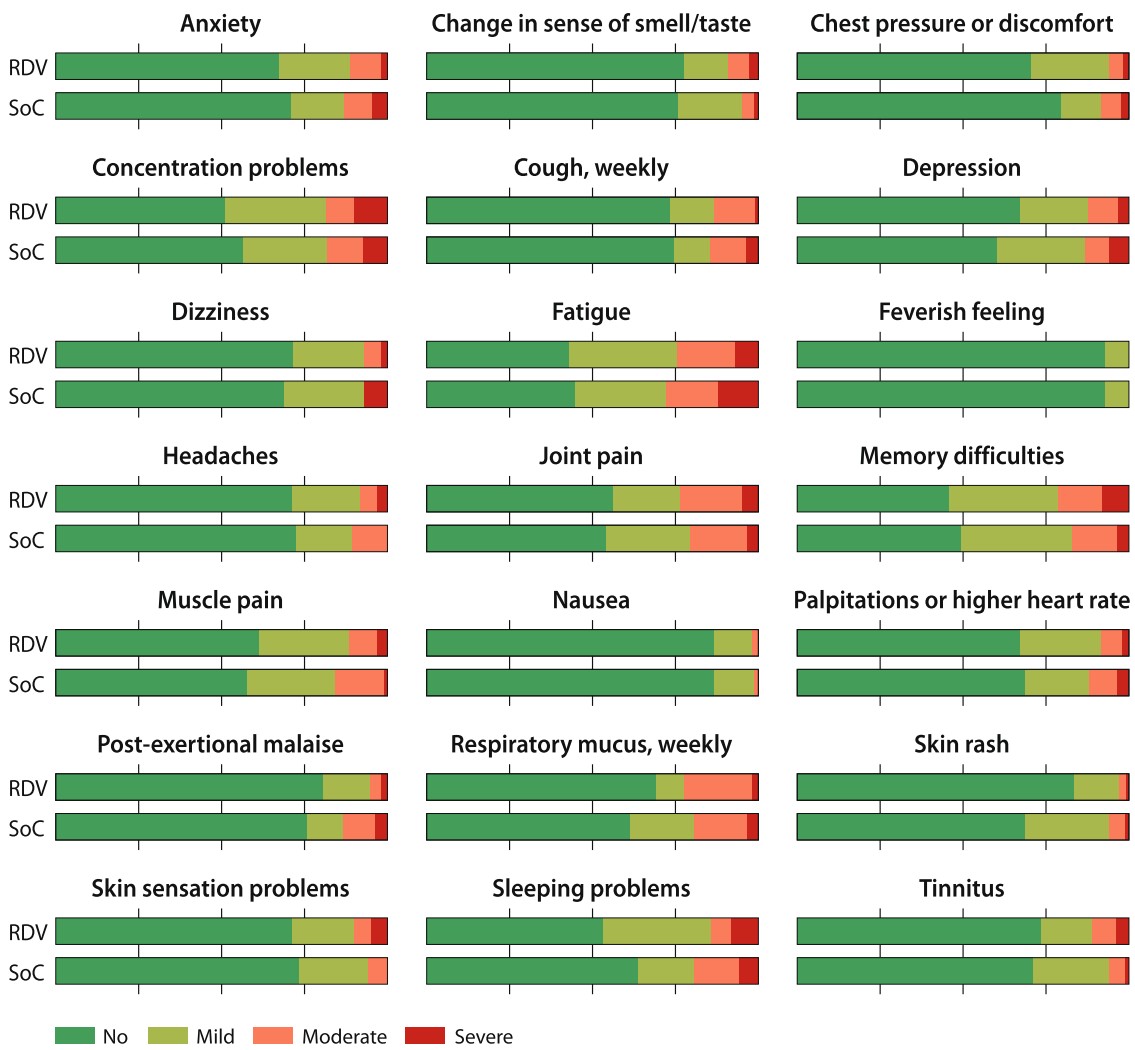

**Fig. 2 | Bother from potential long-COVID symptoms at 1 year from COVID-19 hospitalization between the standard of care and standard of care plus remdesivir groups.** RDV stands for remdesivir plus standard of care group, and SoC for standard of care group.

## Methods

### Study design and patients

We registered the protocol of this follow-up study of the SOLIDARITY Finland trial at ClinicalTrials.gov (NCT04978259). The trial complies with all relevant ethical regulations, and Helsinki University Hospital ethics board approved the study (HUS/1866/2021). This study conforms to the CONSORT 2010 guidance.

We evaluated a range of patient-relevant outcomes 1 year after randomization. The trial was a pragmatic, parallel 1:1 randomized open-label multicenter trial comparing the local standard of care (SoC) and SoC with intravenous remdesivir. We have earlier published the in-hospital (short-term) results of the SOLIDARITY Finland as part of the international WHO SOLIDARITY trial[8,11,15].

We included patients ≥18 years of age with a PCR-confirmed diagnosis of COVID-19 requiring hospitalization (from 11 hospitals in Finland). Patients provided informed consent and were not compensated for participation. We excluded patients who had an estimated life expectancy of <3 months, another acute severe condition during the past week, liver enzyme levels more than five-fold over the upper reference limit, severe kidney failure, or who were pregnant or breastfeeding, or participated in another trial[10]. The WHO SOLIDARITY trial did not perform separate sample size calculations for each participating country[8,15]. The SOLIDARITY Finland recruited patients to the remdesivir trial until the WHO halted the trial[11]. We randomized patients (and collected data) using web-based Castor EDC software

(https://www.castoredc.com). Patients in both groups received SoC. In addition, patients in the remdesivir arm received 200 mg of remdesivir on the first day and 100 mg per day until discharge or for a maximum duration of 10 days.

### Patient outcomes

Our multidisciplinary team of clinicians, methodologists, and patient partners (TR, JS) participated in developing the study design and the questionnaire used to assess long-term recovery (analyses were also stratified by the need of oxygen therapy at randomization) and symptoms (Supplementary). We translated questionnaires, consent forms, and information leaflets from Finnish to Albanian, Arabic, English, Estonian, Persian, Russian, Somalian, and Swedish. We used the modified Medical Research Council dyspnea scale to assess exertional dyspnea, and the EQ-5D-5L and the visual analogue scale (VAS) scale to measure mobility, self-care, usual daily activities, general pain/discomfort, anxiety/depression, and an overall impression of health (Supplementary). We recorded in-hospital deaths in the Castor system, and obtained subsequent death dates up to March 2022 from the Digital and Population Data Services Agency (Helsinki, Finland).

### Statistical analysis

All analyses were unadjusted, intention-to-treat analyses. Statistical significance was defined as an alpha level of 0.05, and

two-sided *p* values were reported. EQ-5D-5L, questions on recovery, and exertional dyspnea had five options ranging from no symptom or problems up to extreme burden from symptom or problems. The questions on potential long-COVID symptoms, including fatigue or problems with attention and concentration, had four response options: no symptom/bother, symptoms with small bother, moderate, and major bother. We made comparisons using the two broad symptom categories (no/small vs. moderate/major bother). We performed all analyses using RStudio version 1.4.1106, and calculated relative risks with 95% CI using the Epi package. The EQ-VAS ranges from 0 (worst imaginable health state) to 100 (best imaginable health state). We analyzed the VAS scale in deciles using ordered logistic regression with the MASS package.

### Reporting summary

Further information on research design is available in the Nature Research Reporting Summary linked to this article.

### Data availability

Detailed aggregate level data are available in the online Supplementary. The dataset generated during and analyzed during the current study are not publicly available for data security. The corresponding author (K.A.O.T.) is the custodian of the data and will provide access to de-identified and processed participant data for academic purposes within 2 months on request (kari.tikkinen@helsinki.fi), with the completion of a data access agreement.

### Code availability

Available in the Supplementary.

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

### Acknowledgements

The authors would like to thank the participating patients, their families, and the hospital staff. The Academy of Finland (335527; K.A.O.T.), Foundation of the Finnish Anti-Tuberculosis Association (K.A.O.T.), Helsinki University Hospital (TYH2022330; K.A.O.T.), Päivikki and Sakari Sohlberg Foundation (K.A.O.T.), Sigrid Jusélius Foundation (K.A.O.T.), Tampere Tuberculosis Foundation (J.R. and K.A.O.T.), and Tampere University Hospital State Research Funding (9AC085; J.R.) funded this study. World Health Organization (WHO) provided the study drug (remdesivir), donated by Gilead Sciences. The funders had no role in the design and conduct of the study; collection, management, analysis, and interpretation of the data; preparation, review, or approval of the manuscript; and the decision to submit the manuscript for publication.

### Author contributions

O.P.O.N., S.H., S.L., J.R., J.M.J.M., and K.A.O.T. designed the study. O.P.O.N. and K.A.O.T. drafted the manuscript. O.P.O.N., J.K.H., G.H.G., and K.A.O.T. performed statistical analyses. O.P.O.N., S.H., S.L., J.R., J.M.J.M., I.E.J.K., H.A., H.-R.K., P.K., J.P., S.P., E.-L.P., M.P., N.P., A.R., Tu.R., Ta.R., J.S., J.K.H., G.H.G., and K.A.O.T. participated to the interpretation of data, and critically revised the manuscript. J.R. and K.A.O.T. obtained funding. O.P.O.N. and K.A.O.T. supervised the study.

### Competing interests

H.-R.K. served on advisory boards for Roche, and Pfizer outside the submitted work. T.R. reported honoraria from AstraZeneca, Boehringer-Ingelheim, and GSK outside the submitted work; and reported having support for attending meetings from MSD, Roche, and Orion outside the submitted work. The remaining authors declare no competing interests.

### Additional information

[1]Faculty of Medicine, University of Helsinki, Helsinki, Finland. [2]Unit of Health Sciences, Faculty of Social Sciences, Tampere University, Tampere, Finland.
[3]Hatanpää Health Center, City of Tampere, Tampere, Finland. [4]Pirkanmaa Hospital District, Tampere, Finland. [5]Department of Internal Medicine, Tampere
University Hospital, Tampere, Finland. [6]Occupational Health Helsinki, City of Helsinki, Helsinki, Finland. [7]Department of Obstetrics and Gynaecology, Helsinki
University and University Hospital Helsinki, Helsinki, Finland. [8]Department of Metabolism, Digestion and Reproduction, Faculty of Medicine, Imperial College
London, London, UK. [9]Research Unit of Clinical Neuroscience, Neurology, University of Oulu, Oulu, Finland. [10]Department of Pulmonology, University of
Helsinki and Helsinki University Hospital, Helsinki, Finland. [11]Faculty of Medicine and Health Technology, Tampere University and Fimlab Ltd., Tampere
University Hospital, Tampere, Finland. [12]Department of Public Health and Welfare, Population Health Unit, Public Health Research Team, Finnish Institute for
Health and Welfare, Helsinki, Finland. [13]Clinical and Molecular Metabolism Research Program, Faculty of Medicine, University of Helsinki, Helsinki, Finland.
[14]Individualized Drug Therapy Research Program, Faculty of Medicine, University of Helsinki, Helsinki, Finland. [15]Department of Pulmonology, Kanta-Häme
Central Hospital, Hämeenlinna, Finland. [16]Suomen Covid -yhdistys ry, Helsinki, Finland. [17]Faculty of Medicine and Health Technology, Tampere University,
Tampere, Finland. [18]Clinicum/Department of Public Health, University of Helsinki, Helsinki, Finland. [19]Department of Health Research Methods, Evidence and
Impact, McMaster University, Hamilton, ON, Canada. [20]Department of Medicine, McMaster University, Hamilton, ON, Canada. [21]Department of Urology,
University of Helsinki and Helsinki University Hospital, Helsinki, Finland. [22]Department of Surgery, South Karelian Central Hospital, Lappeenranta, Finland. *A
list of authors and their affiliations appears at the end of the paper. ✉e-mail: kari.tikkinen@helsinki.fi

## Solidarity Finland Investigators

Tero Ala-Kokko[23], Jaakko Antonen[5], Jutta Delany[24], Heikki Ekroos[25], Riina Hankkio[26], Mia Haukipää[27], Iivo Hetemäki[15],
Pia Holma[28], Ville Holmberg[29], Ville Jalkanen[30], Jenni Jouppila[26], Toni Jämsänen[27], Juuso Järventie[5], Petrus Järvinen[21],
Heikki Kauma[28], Tuomas P. Kilpeläinen[21], Riitta Komulainen[26], Ilari Kuitunen[31,32], Satu M. H. Lamminmäki[33],
Tiina M. Mattila[10], Marjukka Myllärniemi[10], Laura K. Mäkinen[10], Jarkko Mäntylä[10], Gitte Määttä[26], Joni Niskanen[5],
Taina Nykänen[34], Miro Nyqvist[27], Terhi Partanen[28], Riitta-Liisa Patovirta[35], Emmi Puusti[28], Emma Reponen[15], Sari Risku[36],
Mari Saalasti[37], Päivi Salonen[35], Marjatta U. Sinisalo[5], Katariina Sivenius[38], Petrus Säilä[39] & Susanna Tuominen[10]

[23]Division of Intensive Care Medicine, Research Group of Surgery, Anesthesiology, and Intensive Care Medicine, Oulu University Hospital and Medical
Research Center, Oulu, Finland. [24]Accident and Emergency Department, Tampere University Hospital, Tampere, Finland. [25]Department of Pulmonary
Medicine, Porvoo Hospital, Porvoo, Finland. [26]Tampere University Hospital Pharmacy, Tampere, Finland. [27]Department of Pulmonary Medicine, Hyvinkää
Hospital, Hyvinkää, Finland. [28]Research Unit of Biomedicine and Internal Medicine, University of Oulu and Oulu University Hospital, Oulu, Finland.
[29]Department of Infectious Diseases, University of Helsinki and Helsinki University Hospital, Helsinki, Finland. [30]Department of Intensive Care, Tampere
University Hospital, Tampere, Finland. [31]Department of Pediatrics, Mikkeli Central Hospital, Mikkeli, Finland. [32]Institute of Clinical Medicine and Department of
Pediatrics, University of Eastern Finland, Kuopio, Finland. [33]Department of Otorhinolaryngology, Helsinki University Hospital and University of Helsinki,
Helsinki, Finland. [34]Department of Surgery, Hyvinkää Hospital, Hyvinkää, Finland. [35]Division of Respiratory Medicine, Department of Medicine, Kuopio
University Hospital, Kuopio, Finland. [36]Division of Internal Medicine, Seinäjoki Central Hospital, Seinäjoki, Finland. [37]Clinical Trials Unit, HUS Pharmacy,
Helsinki University Hospital, Helsinki, Finland. [38]Division of Infectious Disease, Department of Medicine, Kuopio University Hospital, Kuopio, Finland.
[39]Department of Infectious Diseases, Tampere University Hospital, Tampere, Finland.

