## [Peer review file · Nature Communications]

REVIEWER COMMENTS

Reviewer #1 (Remarks to the Author):

The main result is that remdesivir therapy in acute hospitalised COVID patients does not result in any appreciable difference in long Covid at 1 year. Moreover, there was considerable risk of longer term symptoms at 1 year in all patients post-hospitalised Covid.

The report is consistent with existing literature.

The conclusions are supported by findings.

No flaws in the data analysis or conclusions and sound methodology.

The finding of relatively high prevalence of long COVID 1 year in post-hospitalised COVID patients is not that novel and has been shown in multiple cohorts.

Major: Most of the burden of COVID and Long Covid is non-hospitalised and therefore studying treatments of acute COVID may not be necessarily relevant to those with less severe acute COVID, yet still debilitating Long COVID.

The trial cohort is relatively small compared to other trials and cohorts such as the RECOVERY trial. These trials and those with multiple drugs will have greater value to the scientific community. This is worth acknowledging.

Minor: In the intro, the WHO definition is wrongly quoted as 2 months post-COVID when it is actually 3 months post-COVID.

Reviewer #2 (Remarks to the Author):

The manuscript represents additional, new analyses of the Finnish part of the Solidarity trial of remdesivir versus placebo in hospitalised COVID-19 patients. The authors are commended for performing these long-term follow-up analysis of antiviral treatment in the acute phase of COVID-19.

Due to the small sample size of 181 patients after one year, the authors are correctly being cautious in drawing any firm conclusions.

Some questions need clarifications:

1. Do you have information on time since symptom onset on initiation of remdesivir treatment? Like most antiviral treatment, late initiation of antiviral treatment may reduce its efficacy. Do you have longitudinal data on viral load during treatment of the patients in remdesivir vs placebo? How did the patients in the Finnish sub-cohort do regarding the primary outcomes of the Solidarity trial?
2. Some details are missing in the methods for table 1 and 2. Did you do any adjustment in the RR analysis for age, gender, comorbidity or severity of initial illness? It is known that both severity of disease and gender affect prevalence of long COVID.
3. Outcomes are analysed as individual symptoms. Were any attempts made to look at number of symptoms, if you choose the most prevalent symptoms, in the abstract you mention fatigue, joint pain, memory and concentration problems as the most prevalent.
4. The publication by Boglione et al (doi.org/10.1093/qjmed/hcab297) showing a risk reduction for developing long COVID after remdesivir treatment needs to be cited and discussed.

Reviewer #3 (Remarks to the Author):

This manuscript presents the results of a one-year follow up on symptom and recovery experience in a randomized trial on the administration of remdesivir in individuals hospitalised with Covid-19. There was no observed difference between the two arms of the study for the studied outcome of recovery or reported main symptoms after one year.

Despite the relatively small sample size, well acknowledged in the discussion, the paper is a good starting point for the community to look into.

The design of the study is well presented but there seem to be a minor discrepancy in numbers: the authors say they have received 181 answers to their questionnaire at follow-up but only list 26 individuals missing (5 who died during hosp, 5 during the following year, 11 who were not reachable and 5 who refused to participate) which would put the number to 182. It seems from the flowchart in supplementary material that 6 individuals died during the year of follow-up). Have the reasons for not being able to reach the individuals be further investigated (hospitalisation, admitted to care homes, having to move in with relatives...). As this could be a marker of recovery issue, it may be important to mention it in the discussion.

It is not very clear from the manuscript what was used as covariate in the analysis (were sex, BMI and age corrected for in the statistical analyses?)

It would make sense to also present the proportion of individuals still experiencing any symptom after 1 year so as to compare with other published papers on outcome and follow-up after hospitalisation especially as the proportion of individuals still experiencing any symptom after 6 months appears to be in the 70%. Some discussion about it would be interesting.

The authors mention the time at which remdesivir was started and how long it lasted but there does not seem to be any analysis of the effect of either treatment duration or time from hospitalisation on the outcome here. Such an experiment would be an interesting possible addition. One point of discussion that would be needed or analysis if possible would be to know whether people received treatment after discharge from hospital and if so, when it stopped.

Effect of remdesivir on recovery, quality of life, and long-COVID symptoms one year after hospitalization for COVID-19 infection: the randomized controlled SOLIDARITY Finland trial

Manuscript ID: NCOMMS-22-21863-T

RESPONSE TO PEER REVIEW

Please find below our point-by-point responses to the comments raised by the reviewers. For each comment, we responded with the following:

- 1) Our response to the comment
 - 2) How this resulted in a change in the manuscript (or did not)
 - 3) Where the change(s) in the revised manuscript can be found
-

COMMENTS FROM EXTERNAL PEER REVIEWERS

Reviewer #1

Comment #1: *“The main result is that remdesivir therapy in acute hospitalised COVID patients does not result in any appreciable difference in long Covid at 1 year. Moreover, there was considerable risk of longer term symptoms at 1 year in all patients post-hospitalised Covid.”*

The report is consistent with existing literature. The conclusions are supported by findings. No flaws in the data analysis or conclusions and sound methodology. The finding of relatively high prevalence of long COVID 1 year in post-hospitalised COVID patients is not that novel and has been shown in multiple cohorts.”

REPLY:

- 1) Thank you for these positive comments.
- 2) No changes required.

Comment #2: *“Most of the burden of COVID and Long Covid is non-hospitalised and therefore studying treatments of acute COVID may not be necessarily relevant to those with less severe acute COVID, yet still debilitating Long COVID.”*

REPLY:

- 1) Thank you for the comment. We agree that results from our hospital study population provide less direct evidence for non-hospitalized patients.
- 2) We have revised the limitations section as follows: *“As the short-term benefits of remdesivir seem larger in non-severe than in critically ill patients, this could be the subpopulation of hospitalized patients who might also achieve long-term benefits. Patients experienced much lower in-hospital mortality rates in Finland (2.4%) than in the global trial (15.0%) and were therefore a potentially more suitable patient population (likely*

earlier phase of the disease) for an antiviral drug. However, our subgroup of patients in this phase was too small to inform this hypothesis. We hope that other COVID-19 treatment RCTs, especially the large trials – such as the RECOVERY and PANORAMIC trials - will also extend their follow-up to evaluate the potential long-term effects of acute-phase treatment both in hospitalised and non-hospitalised patients.”

3) Pages 6-7

Comment #3: *“The trial cohort is relatively small compared to other trials and cohorts such as the RECOVERY trial. These trials and those with multiple drugs will have greater value to the scientific community. This is worth acknowledging.”*

REPLY:

- 1) Thank you for the comment. We have now emphasized more the importance of larger trials in the discussion.
- 2) Revised text is as follows: *“However, our subgroup of patients in this phase was too small to inform this hypothesis. We hope that other COVID-19 treatment RCTs, especially the large trials – such as the RECOVERY and PANORAMIC trials - will also extend their follow-up to evaluate the potential long-term effects of acute-phase treatment both in hospitalised and non-hospitalised patients.”*
- 3) Page 6-7.

Comment #4: In the intro, the WHO definition is wrongly quoted as 2 months post-COVID when it is actually 3 months post-COVID.

REPLY:

- 1) Thank you for correcting this mistake.
- 2) In the Intro it is now re-phrased as *“The World Health Organization (WHO) defines long COVID as symptoms continuing beyond three months from the onset of COVID-19 without an alternative explanation”*
- 3) Page 4

Reviewer #2

Comment #1: *“The manuscript represents additional, new analyses of the Finnish part of the Solidarity trial of remdesivir versus placebo in hospitalised COVID-19 patients. The authors are commended for performing these long-term follow-up analysis of antiviral treatment in the acute phase of COVID-19. Due to the small sample size of 181 patients after one year, the authors are correctly being cautious in drawing any firm conclusions.”*

REPLY:

- 1) Thank you for these comments.
- 2) No changes required.

Comment #2: *“Do you have information on time since symptom onset on initiation of remdesivir treatment? Like most antiviral treatment, late initiation of antiviral treatment may reduce its efficacy. Do you have longitudinal data on viral load during treatment of the patients in remdesivir vs placebo? How did the patients in the Finnish sub-cohort do regarding the primary outcomes of the Solidarity trial?”*

REPLY:

1) Thank you for these comments. SOLIDARITY was conducted as a pragmatic trial within a pandemic that challenged the resources of the health care system. Within this context, the SOLIDARITY Finland trial recorded only the most necessary information at the time of hospitalization. The SOLIDARITY trial did not record data on how many days before the symptoms had started, or information on viral load. We did, however, record when remdesivir was started in relation to the beginning of the hospitalization. In the SOLIDARITY Finland trial, we started remdesivir in 65 patients (58%) on the hospital admission day or the day after, in 36 patients (32%) on the second day after admission, in 10 patients (9%) on the third day after hospital admission, and two (2%) later. The median duration of remdesivir treatment was 5 days (interquartile range IQR 4–8).

In-hospital mortality in SOLIDARITY Finland was low (5/208 patients, 2.4 %) compared to the international WHO SOLIDARITY trial (1245/8275, 15 %), reflecting the Finnish hospitals recruitment of patients at an earlier phase of the disease.

2) Prompted by these comments, we revised the text as follows:

i) The revised Results text is as follows: *“Remdesivir was started on either the day of hospital admission or the first full day of hospitalization in 65 patients (57%), on the second full day of hospitalization in 36 patients (32%), and later in 12 patients (11%).”*

ii) When discussing the impact of timing of remdesivir, we also acknowledge the mortality difference between Solidarity Finland and WHO Solidarity, prompted by these comments, as follows: *“Patients experienced much lower in-hospital mortality rates in Finland (2.4%) than in the global trial (15.0%), and were therefore a potentially more suitable patient population (likely earlier phase of the disease) for an antiviral drug. However, our subgroup of patients in this phase was too small to inform this hypothesis.”*

3) Pages 4 and 6.

Comment #3: *“Some details are missing in the methods for table 1 and 2. Did you do any adjustment in the RR analysis for age, gender, comorbidity or severity of*

initial illness? It is known that both severity of disease and gender affect prevalence of long COVID.”

REPLY:

- 1) Thank you for these comments. We agree that it is a good idea to add the missing baseline numbers of five hospital phase variables also for the 1-year follow-up participants.
We performed a randomized trial with similar baseline age, gender and severity of initial illness between remdesivir and standard of care arms. Our outcomes included quality of life (EQ-5D-5L), recovery, exertional dyspnea, and occurrence/bother from potential long COVID symptoms. We performed a pre-specified subgroup analysis by COVID severity (oxygen need) but, as anticipated due to the small sample size, we could not detect effect modification. All our analyses were non-adjusted.
- 2) We have now added the missing baseline numbers for the 1-year follow-up participants to the Table 1.
We revised the first sentence of the statistical analysis as follows: *“All analyses were unadjusted, intention-to-treat analyses.”*
- 3) Table 1 and page 9.

Comment #4:

“Outcomes are analysed as individual symptoms. Were any attempts made to look at number of symptoms, if you choose the most prevalent symptoms, in the abstract you mention fatigue, joint pain, memory and concentration problems as the most prevalent.”

REPLY:

- 1) Thank you for the comment. We ran statistical analyses only for pre-specified symptoms. In Figure 2, we also show the distribution of all symptoms between the treatment arms. Looking at a combination of symptoms or grouping them would be a post hoc analysis, which we would prefer not to perform as part of this paper (as recommended by international guidelines on trial methodology). We are happy to reconsider this if Editors see required.
- 2) No changes.

Comment #5:

“The publication by Boglione et al (doi.org/10.1093/qjmed/hcab297) showing a risk reduction for developing long COVID after remdesivir treatment needs to be cited and discussed.”

REPLY:

- 1) Thank you very much for this comment. We agree that it is good idea to cite and discuss the paper by Boglione and coworkers.
- 2) We have revised the Introduction as follows: *“An observational study among more than 400 hospitalized patients in Italy suggested lower risk of long-COVID for those who received remdesivir during hospitalization (OR 0.64, 95% CI 0.41-0.78). There are, however, no published long-term follow-up studies of randomized controlled trials (RCTs) of COVID-19 treatments.”*

We have also revised the Discussion as follows: *“This is the first long-term follow-up of an RCT on COVID-19 drug treatment. We could not demonstrate long-term benefits for remdesivir in patients hospitalized due to COVID-19. After a one-year follow-up, one in six survivors reported they had not recovered well from COVID-19. One in four reported substantial bother from fatigue. This result is similar that reported in an Italian observational study of hospitalised patients that reported fatigue in one in three patients at 6 months post COVID-19.”*

3) Pages 4 and 6.

Reviewer #3

Comment #1: *“This manuscript presents the results of a one-year follow up on symptom and recovery experience in a randomized trial on the administration of remdesivir in individuals hospitalised with Covid-19. There was no observed difference between the two arms of the study for the studied outcome of recovery or reported main symptoms after one year.*

Despite the relatively small sample size, well acknowledged in the discussion, the paper is a good starting point for the community to look into.”

REPLY:

- 1) Thank you for these positive comments.
- 2) No change.

Comment #2: *“The design of the study is well presented but there seem to be a minor discrepancy in numbers: the authors say they have received 181 answers to their questionnaire at follow-up but only list 26 individuals missing (5 who died during hosp, 5 during the following year, 11 who were not reachable and 5 who refused to participate) which would put the number to 182. It seems from the flowchart in supplementary material that 6 individuals died during the year of follow-up). Have the reasons for not being able to reach the individuals be further investigated (hospitalisation, admitted to care homes, having to move in with relatives...). As this could be a marker of recovery issue, it may be important to mention it in the discussion.”*

REPLY:

- 1) Thank you for this comment. Up to one year, there were a total of 10 deaths. Subsequently, one patient died at 417 days. The original flowchart in the supplement (but not the main text) incorrectly showed deaths beyond one year, and therefore, this 11th death (6th in the SoC group) was shown. We did not receive a response from this patient before the death.
- 2) We have now revised the flow chart and classified the patient as not reached in the flow chart. This information (the flow chart) is now as the Figure 1 rather than in the Supplement.
- 3) Figure 1.

Comment #3: *“It is not very clear from the manuscript what was used as covariate in the analysis (were sex, BMI and age corrected for in the statistical analyses?)”*

REPLY:

- 1) Thank you for your comment. The statistical analyses were unadjusted. Please see our response to Reviewer #1 Comment #3 for further details.
- 2) We revised the first sentence of the statistical analysis as follows: *“All analyses were unadjusted, intention-to-treat analyses.”*
- 3) Page 9.

Comment #4: *“It would make sense to also present the proportion of individuals still experiencing any symptom after 1 year so as to compare with other published papers on outcome and follow-up after hospitalisation especially as the proportion of individuals still experiencing any symptom after 6 months appears to be in the 70%. Some discussion about it would be interesting.”*

REPLY:

- 1) Thank you for your comment. Our study is an RCT, and according to the international guidance regarding trials, we would prefer to stay with the predefined outcomes. Please see our response to Reviewer #2 Comment #4 for further details. We are happy to re-consider this if Editors see this as required.
- 2) No changes.

Comment #5: *“The authors mention the time at which remdesivir was started and how long it lasted but there does not seem to be any analysis of the effect of either treatment duration or time from hospitalisation on the outcome here. Such experiment would be an interesting possible addition. One point of discussion that would be needed or analysis if possible would be to know whether people received treatment after discharge from hospital and if so, when it stopped.”*

REPLY:

- 1) Thank you very much for your comments and review. Due to the limited sample size, the only subgroup analysis we performed, and which was predetermined, was by COVID-19 severity. Additional subgroup analyses with a small dataset, particularly post hoc analyses, would be inadvisable. We address, in the revised discussion, whether remdesivir’s effect depends on the timing of administration. We do not have information regarding potential treatments post-discharge. However, as no treatment has been approved for Long-COVID, this may not be a major limitation. We have added this as limitations to the text.
- 2) We revised the discussion regarding the timing of the remdesivir and COVID-19 as follows: *“Adequately powered studies (with short follow-up) have earlier demonstrated that remdesivir can prevent the progression to*

severe disease, if given in the early phase of the infection.^{8,10} A fifth of our patients were not receiving oxygen at hospital admission. As the short-term benefits of remdesivir seem larger in non-severe patients⁹, this could be the subpopulation of hospitalized patients who might also achieve long-term benefits. *Patients experienced much lower in-hospital mortality rates in Finland (2.4%) than in the global trial (15.0%) and were therefore a potentially more suitable patient population (likely earlier phase of the disease) for an antiviral drug.* However, our subgroup of patients in this phase was too small to inform this hypothesis.”

We also revised the end of the limitations as follows: “*Finally, we did not have information regarding potential treatments post-discharge. However, as no specific treatment has been proved effective for Long-COVID, this may not be a major limitation.*”

3) Pages 6 and 7.